EMBO
Molecular Medicine

# *In vivo* generation of human CD19-CAR T cells results in B-cell depletion and signs of cytokine release syndrome

Anett Pfeiffer[1,†], Frederic B Thalheimer[1,†], Sylvia Hartmann[2], Annika M Frank[1], Ruben R Bender[1], Simon Danisch[3], Caroline Costa[4], Winfried S Wels[5,6,7], Ute Modlich[8], Renata Stripecke[3], Els Verhoeyen[4,9] & Christian J Buchholz[1,7,10,*] (iD)

## Abstract

Chimeric antigen receptor (CAR) T cells brought substantial benefit to patients with B-cell malignancies. Notwithstanding, CAR T-cell manufacturing requires complex procedures impeding the broad supply chain. Here, we provide evidence that human CD19-CAR T cells can be generated directly *in vivo* using the lentiviral vector CD8-LV specifically targeting human CD8⁺ cells. Administration into mice xenografted with Raji lymphoma cells and human peripheral blood mononuclear cells led to CAR expression solely in CD8⁺ T cells and efficacious elimination of CD19⁺ B cells. Further, upon injection of CD8-LV into mice transplanted with human CD34⁺ cells, induction of CAR T cells and CD19⁺ B-cell depletion was observed in 7 out of 10 treated animals. Notably, three mice showed elevated levels of human cytokines in plasma. Tissue-invading CAR T cells and complete elimination of the B-lymphocyte-rich zones in spleen were indicative of a cytokine release syndrome. Our data demonstrate the feasibility of *in vivo* reprogramming of human CD8⁺ CAR T cells active against CD19⁺ cells, yet with similar adverse effects currently notorious in the clinical practice.

**Keywords** cytokine release syndrome; gene delivery; humanized mouse; T-cell targeting

**Subject Categories** Cancer; Immunology

See also: **M Feldmann** (November 2018)

## Introduction

The genetic modification of T cells with chimeric antigen receptors (CARs) recognizing surface antigens on malignant cells has become a valid immune therapeutic option for cancer patients (Lim & June, 2017). More than 200 clinical trials assessing *ex vivo*-generated CAR T cells have been initiated in recent years (Chmielewski & Abken, 2015; Hartmann *et al*, 2017). Especially, for the treatment of pre-B and B-cell malignancies remarkable responses have been obtained with CD19-directed CARs, which have recently been granted marketing approval for two products by the FDA (Riviere & Sadelain, 2017). Intense efforts are ongoing to extend the CAR T-cell approach to other liquid and solid tumors and debilitating viral infections (Irving *et al*, 2017).

Chimeric antigen receptors are composed of an antigen binding domain, which most often consists of a single-chain antibody fragment (scFv), hinge and transmembrane domains, and intracellular signaling domains derived from the CD3ζ chain and CD28 or 4-1BB. Upon antigen recognition and binding, CAR T cells become activated, expand, and kill target cells. After depletion of the target cells and due to intrinsic signaling characteristics, the bulk levels of CAR T cells decline. Nevertheless in some clinical trials, CD19-CAR T cells persisted functionally for years in patients (Porter *et al*, 2015).

Chimeric antigen receptor T cells are regarded as individualized cell therapy products. Production starts with the patients' T cells, which are genetically modified and expanded *ex vivo* before finally being re-infused (Levine *et al*, 2017). Stable expression of CARs is essential to maintain the cells functional and persistent.

1   Molecular Biotechnology and Gene Therapy, Paul-Ehrlich-Institut, Langen, Germany
2   Dr. Senckenberg Institute of Pathology, Goethe-University Frankfurt, Frankfurt am Main, Germany
3   Department of Hematology, Hemostasis, Oncology and Stem Cell Transplantation, Laboratory of Regenerative Immune Therapies Applied, Excellence Cluster REBIRTH and German Centre for Infection Research (DZIF), partner site Hannover, Hannover, Germany
4   CIRI – International Center for Infectiology Research, Team EVIR, Inserm, U1111, CNRS, UMR5308, Ecole Normale Supérieure de Lyon, Université Claude Bernard Lyon 1, University of Lyon, Lyon, France
5   Institute for Tumor Biology and Experimental Therapy, Georg-Speyer-Haus, Frankfurt, Germany
6   German Cancer Consortium (DKTK), partner site Frankfurt/Mainz, Frankfurt, Germany
7   German Cancer Research Center (DKFZ), Heidelberg, Germany
8   Division of Veterinary Medicine, Research Group for Gene Modification in Stem Cells, Paul-Ehrlich-Institut, Langen, Germany
9   INSERM, C3M, Université Côte d'Azur, Nice, France
10  German Cancer Consortium (DKTK), partner site Heidelberg, Heidelberg, Germany
*Corresponding author. Tel: +49 6103774011; E-mail: christian.buchholz@pei.de
†These authors contributed equally to this work as first authors

Accordingly, lentiviral (LV) or γ-retroviral vectors are currently the most frequently employed gene modification systems in the clinics (Schambach & Morgan, 2016). While the therapeutic concept has been validated, the manufacturing process and supply chain are highly demanding and resource-intensive. A straight-forward transfer of the current manufacturing process into routine clinical practice, especially considering the high number of cancer patients, appears extremely difficult, if not impossible to implement (Hartmann et al, 2017). Therefore, efforts to generate allogeneic CAR T cells which are compatible for transplantation into HLA-matched patients are ongoing (Qasim et al, 2017). However, even if this can be achieved, CAR T cells will still remain a complex product to manufacture. Alternatively, in situ reprogramming of cytotoxic CD8+ CAR T cells through direct injection of the gene vector could dramatically bypass these limitations.

Efficient and highly selective gene delivery into T cells in vivo represents a particular challenge in achieving this goal. Besides selectivity, also the usually resting state of T cells in vivo which is not compatible with gene delivery by conventional LVs poses a problem (Amirache et al, 2014). Precise in vivo gene delivery into distinct cell types of choice has been achieved through targeting of LVs to recognize distinct surface markers as entry receptors (Anliker et al, 2010; Buchholz et al, 2015). Recently, we reported on the engineering of human CD4- (CD4-LV) and CD8-targeted vectors (CD8-LV) capable to deliver genes selectively into human CD4+ or CD8+ T cells (Zhou et al, 2012, 2015; Bender et al, 2016). Toward in situ generation of CAR T cells, here we report that CD19-reactive CD8+ CAR T cells can be generated in humanized mice upon a single systemic administration of CD8-LV. As envisioned, in vivo CAR T-cell reprogramming was accompanied by selective B-cell depletion. Notably, some of the animals developed symptoms reminiscent of the cytokine release syndrome (CRS) sporadically observed in CAR T-cell-treated patients (Hay et al, 2017).

## Results and Discussion

To test the hypothesis that CD19-CAR T cells can be directly generated in vivo, we packaged the genetic information for a CD19-CAR harboring the FMC63 scFv and the CD28ζ signaling domain into CD8-LV. Upon in vitro transduction of human PBMC, CAR expression was selectively detectable in CD8+ T cells (Figs 1A and EV1A). These cells killed CD19+ B cells and Raji cells but not CD19− control cells (Fig EV1B and C). To assess this vector for the reprogramming of CAR T cells in vivo, NOD-scid-IL2Rγ$^{null}$ (NSG) mice were transplanted with activated human PBMC. Since activated PBMC contain only very few B lymphocytes, which do not engraft well, Raji lymphoma cells were transplanted in addition to serve as target and provide antigenic stimulation for in vivo-induced CAR T cells (Fig 1B). After injection of the vector particles, higher frequencies of CD8+ T cells were present in CD8-LV$^{CD19CAR}$-injected mice versus control mice, especially in the peritoneal cavity (Fig 1D and Appendix Fig S1). Among the CD8+ cells of vector-injected mice, the frequencies of CAR T-positive cells were 30–50% in peritoneal fluid, 10–35% in spleen, and 5–30% in blood (Fig 1E and Appendix Fig S2). These high numbers of CAR+ T cells were well in agreement with the vector copy numbers (VCN) present in genomic DNA isolated from the tissues (Figs 1C and EV2). Since there was no overall increase in human CD45+ cells (Appendix Fig S1) and in vivo transduction rates with the reporter gene encoding vector CD8-LV$^{RFP}$ remained below 5%, this must have been due to preferential proliferation of the initially transduced cells (Fig 1E). Notably, less than 0.5% of the CD8− cells were detected in the CAR+ gate (Fig 1E). Remarkably, all mice that had received CD8-LV$^{CD19CAR}$ essentially lacked human CD19+ cells in peritoneal cavity, spleen, and blood (Fig 1F). Since control mice contained low but significantly higher frequencies of CD19+ cells, they must have been eliminated by the in vivo-generated CD19-CAR T cells (Fig 1F). The peritoneal fluid, blood, and spleen mainly contained human B cells, while the Raji cells had rather spread into the peritoneal tissue (Appendix Fig S3). To test whether residual CD19+ B cells present in the human PBMC had activated the in vivo-generated CAR T cells, we injected the mice with CD19+ cell-depleted PBMC prior to vector administration. In this setting, only about 5% CAR+ cells were detected in the peritoneal fluid, while signals in blood and spleen were close to background (Fig 1G). This signifies that even low frequencies of CD19+ B cells present in PBMC were sufficient to activate CAR+/CD8+ T cells and promote their expansion.

In agreement with that, we found a clear polyclonal situation for the CAR+ cells of the peritoneal cavity in the presence of target cells, while in their absence or upon RFP gene transfer, the transduced cells were more oligoclonal, since then distinct bands were

---

**Figure 1. CAR T-cell generation in human PBMC-transplanted mice.**

A    *Ex vivo* generation of CAR T cells. Activated human PBMC were left untransduced or incubated with CD8-LV$^{CD19CAR}$ at an MOI of 2. Five days later, expression of CD19-CAR and CD8 was determined on CD3+ cells. Numbers indicate the percentage of cells in the respective gate.

B    Experimental outline for *in vivo* CAR generation. 1 × 10$^7$ human PBMC were engrafted into naïve NSG mice or NSG mice that had been intraperitoneally (i.p.) injected with 5 × 10$^5$ Raji cells (Raji+) 6 days before. One day later, 2 × 10$^6$ t.u. of CD8-LV$^{CD19CAR}$ (filled circles) or CD8-LV$^{RFP}$ (gray triangles) were i.p. injected, respectively. As further control, another group of mice received PBS (open circles). Seven days later, mice were sacrificed and organs and cells were removed for further analysis.

C    Detection of CAR T cells by vector copy numbers (VCN). Genomic DNA was isolated from peritoneal cavity, spleen, and blood cells. VCN were determined in technical duplicates by qPCR for two individual mice of each group. The presence of B cells in the transplanted PBMC is indicated below.

D–F    Cells isolated from the peritoneal cavity (peritoneum), spleen, or blood were evaluated by flow cytometry for the percentages of human CD8+ in CD3+ cells (D), of CAR+ or RFP+ cells in the CD8+ and CD8− fractions, respectively (E), and of human CD19+ cells (F) within the fraction of human CD45+ cells. Representative density plots are shown for the peritoneal cells. The gating strategy is represented in Appendix Fig S1A.

G    Mice were transplanted with B-cell-depleted PBMC and then received CD8-LV$^{CD19CAR}$ (filled circle) or PBS (open circle). As control, CD8-LV$^{CD19CAR}$ or PBS was injected into mice transplanted with non-depleted PBMC.

Data information: Data represent mean ± SD for all groups (CD8-LV$^{CD19CAR}$: n = 6 (+Raji) and n = 4 (−Raji) in (D), n = 4 (−B-cells) in (G); CD8-LV$^{RFP}$: n = 4; PBS: n = 4 in (G), all others n = 3). Statistical significance was determined using Mann–Whitney test; ns, not significant.

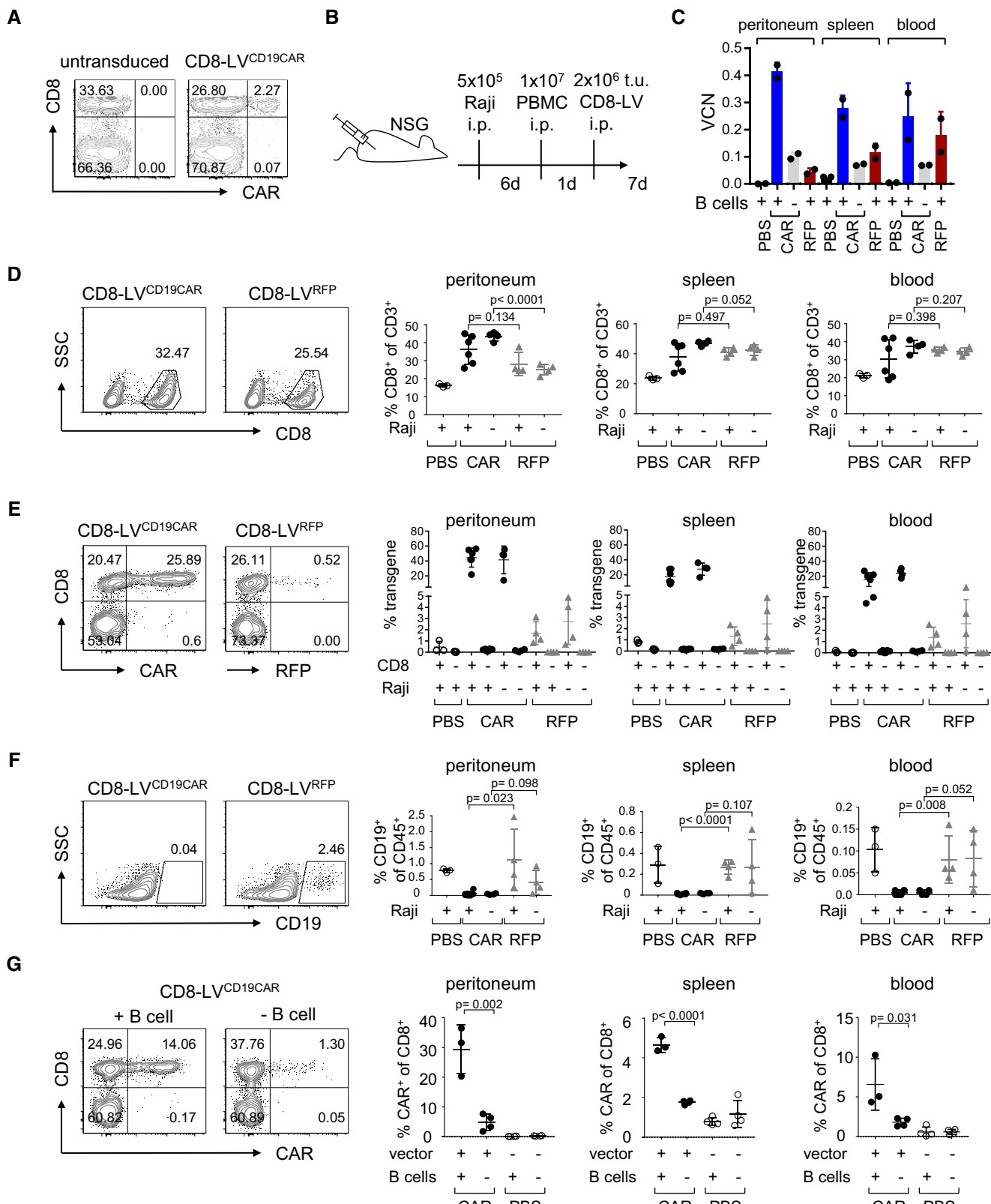

**Figure 1.**

detected by LM-PCR. In spleen, a more oligoclonal pattern was present (Fig EV3A). PD-1 expression was high in the T cells of all mouse groups, which is in line with the overall activation of the transplanted PBMC (Fig EV3B). Notably, vector-injected mice exhibited even higher PD-1 levels, which is likely due to the stimulating activity of the CD8-specific scFv displayed on the particle surface (Zhou *et al*, 2012). The activation/exhaustion markers LAG-3 and TIM-3, in contrast, were selectively upregulated in CAR$^+$ T cells of the peritoneal cavity. This difference was much less pronounced in spleen (Fig EV3B). These data are well in agreement with a preferential expansion of CAR$^+$ cells in the peritoneal cavity in the presence of antigen followed by migration of a fraction of less exhausted CAR T cells to spleen.

These data provided proof of principle that human CD19-CAR T cells could be generated *in vivo* using LVs targeted to CD8. CAR expression was thereby restricted to human CD8$^+$ T cells. *In vivo*-generated CAR T cells expanded upon recognition of CD19$^+$ cells in the peritoneal cavity, eliminated CD19$^+$ cells, and migrated to lymphatic tissues. This human PBMC xenograft model was not physiological due to the intrinsic high activation of the adoptive T cells (expressing CAR or not) resulting in graft-versus-host disease (GvHD) and restricting the observation period. In addition, this overt T-cell activation renders T cells more permissive to LV transduction and, importantly, confers higher expression levels of the transferred gene than in endogenously developed resting T cells. This holds true also for expression from the SFFV promoter in T cells (Frecha *et al*, 2008), which we used here.

To assess whether CAR T cells could also be generated from T cells in steady-state, vector particles were injected into mice transplanted with human CD34$^+$ hematopoietic stem cells (HSCs). In these animals, human T cells develop in the mouse thymus and migrate and mature in the secondary lymphatic tissues (Walsh *et al*, 2017). Humanized mice contained 38–72% human CD45$^+$ cells in blood, of which 24–87% were human CD19$^+$ B cells and 2–26% human CD8$^+$ cells (see Appendix Fig S4 for details and group allocation). These mice were administered either with PBS or with CD8-LV$^{CD19-CAR}$. Both groups received intravenous injections with interleukin-7 (IL-7) prior to vector administration for homeostatic activation of the CD8$^+$ T cells (Verhoeyen *et al*, 2003; Fig 2A). IL-7 is a homeostatic cytokine, which promotes cell viability and pushes resting T lymphocytes into the G1B phase of the cell cycle making them more permissive for lentiviruses (Cavalieri *et al*, 2003; Verhoeyen *et al*, 2003; Swainson *et al*, 2006), and has been safely applied to patients in clinical studies (Rosenberg *et al*, 2006). Seven

to 18 weeks after vector administration, CAR T cells were detected in the blood of six out of 10 mice as determined by flow cytometry. Notably, CAR expression was detectable exclusively in the human CD8$^+$ T-cell population ranging between 1.5 and 14% of all CD8$^+$ lymphocytes (Fig 2B). One animal was definitely devoid of CAR T cells. Three additional mice possibly contained CAR T cells, but signals were not clearly above the detection limit. PCR performed on genomic DNA isolated from bone marrow of all animals was overall in agreement with flow cytometry results and revealed in total seven vector-injected mice being CAR-positive by PCR (Fig 2C and D). CAR gene copy numbers were in most animals higher than expected from the flow cytometry data. Multiple integrations in single cells as well as inactivation of the SFFV promoter, which has been previously observed (Stein *et al*, 2010), could offer possible explanations for this discrepancy. Moreover, some *in vivo*-generated CAR T cells may have returned to a resting state after the transient IL-7 stimulation. Such resting or minimally activated T cells may not express the CAR at sufficiently high levels to be detected by flow cytometry.

Flow cytometry of spleen and bone marrow cells was in agreement with the values observed in blood. With up to 15% CAR$^+$ cells among the human CD8$^+$ cells, animals M16 and M19 contained the highest CAR T-cell levels of all mice in these tissues (colored symbols in Fig 2B). In contrast, CAR$^+$ cells were virtually absent within the CD8$^-$ cells of all vector-injected animals which thus did not significantly deviate from the PBS group (Fig 2B). Notably, the high CAR T-cell levels in M16 and M19 were accompanied by complete absence of CD19$^+$ lymphocytes in blood, spleen, and bone marrow (Fig 2E). Also when all CAR$^+$ mice were related to the levels before vector injection for each mouse individually, CD19$^+$ levels had significantly decreased (Fig 2F). No reduction in CD19$^+$ cells was detected in any organ of the three CAR-negative (CAR$^-$) mice (Fig 2E and F). Animal M16 had to be sacrificed about 7 weeks after vector injection due to weight loss, ruffled fur, apathy, ataxia, and moving in circles, and mouse M19 reached endpoint at day 53. Both mice showed splenomegaly and hypocellular bone marrow. Due to abnormal behavior, weight loss, and apathy, two additional mice were sacrificed 10 weeks after vector administration.

In order to assess whether a CRS pattern could be associated with these symptoms, we measured the concentrations of 12 different human cytokines in the mouse plasma. Three CAR$^+$ mice, among these also M16 and M19, had substantially elevated levels of certain cytokines (Figs 3A and EV3). Inflammation-related cytokines such as IL-6, interferon-γ (IFNγ), and granulocyte-macrophage

**Figure 2. CAR T-cell generation in HSC-transplanted mice.**

A  Experimental outline. IL-7 was injected intravenously into HSC-humanized NSG mice before 2 × 10$^6$ t.u. of CD8-LV$^{CD19CAR}$ (CAR), or PBS as control (PBS), were administered.

B  CAR T-cell levels in the CD8$^+$ and CD8$^-$ T cells harvested from blood, spleen, and bone marrow of PBS (PBS) or vector-injected (CAR) mice determined by flow cytometry.

C  Detection of CAR T cells by determining vector copy numbers (VCN) in genomic DNA isolated from CD8$^+$-enriched cells harvested from bone marrow.

D  Detection of CAR T cells in bone marrow quantified by flow cytometry.

E  CD19$^+$ B-cell levels in blood, spleen, and bone marrow determined by flow cytometry.

F  Relative human CD19$^+$ B-cell level in blood calculated by normalizing the levels at the day the animals were sacrificed to those before vector administration.

Data information: Distinct symbols for each individual animal are used throughout panels (B–F) (M16: lilac triangle; M19: blue triangle). Open symbols indicate animals from the vector-injected group devoid of PCR-detectable CAR T cells. Data represent mean ± SD for all groups. *N* = 6 in PBS group; *n* = 10 in CAR group. Statistical significance was determined using one-way ANOVA test with Bonferroni correction.

colony-stimulating factor (GM-CSF) were the most prominent. Tumor necrosis factor-α (TNFα) was increased in M16 and M19, and there was a tendency toward elevated IL-10, IL-2, and IL-9 in

some of the vector-injected mice (Fig 3A). In order to confirm the CRS pathology in animals M16 and M19, histological sections from various organs were evaluated. Infiltrating lymphocytes were

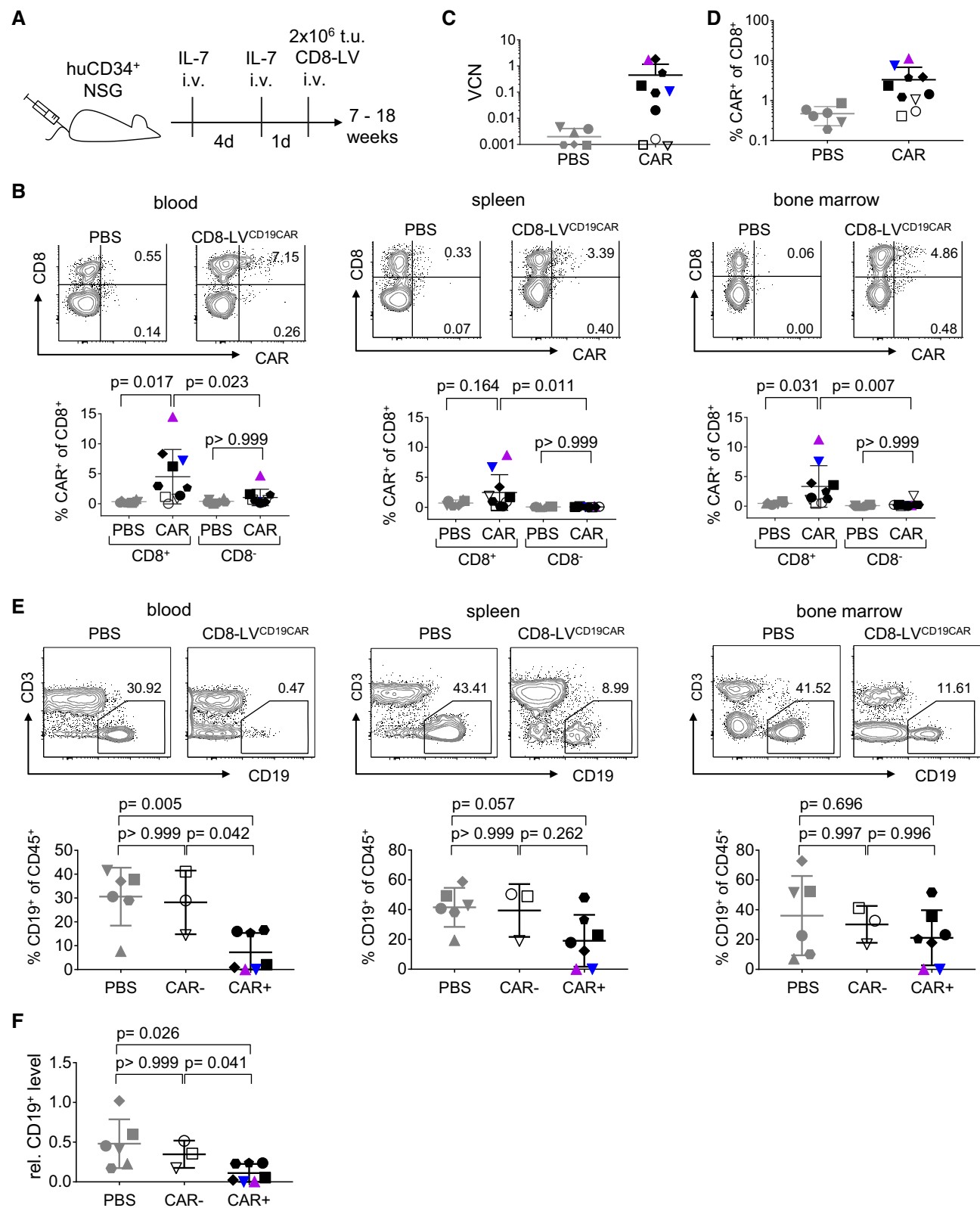

Figure 2.

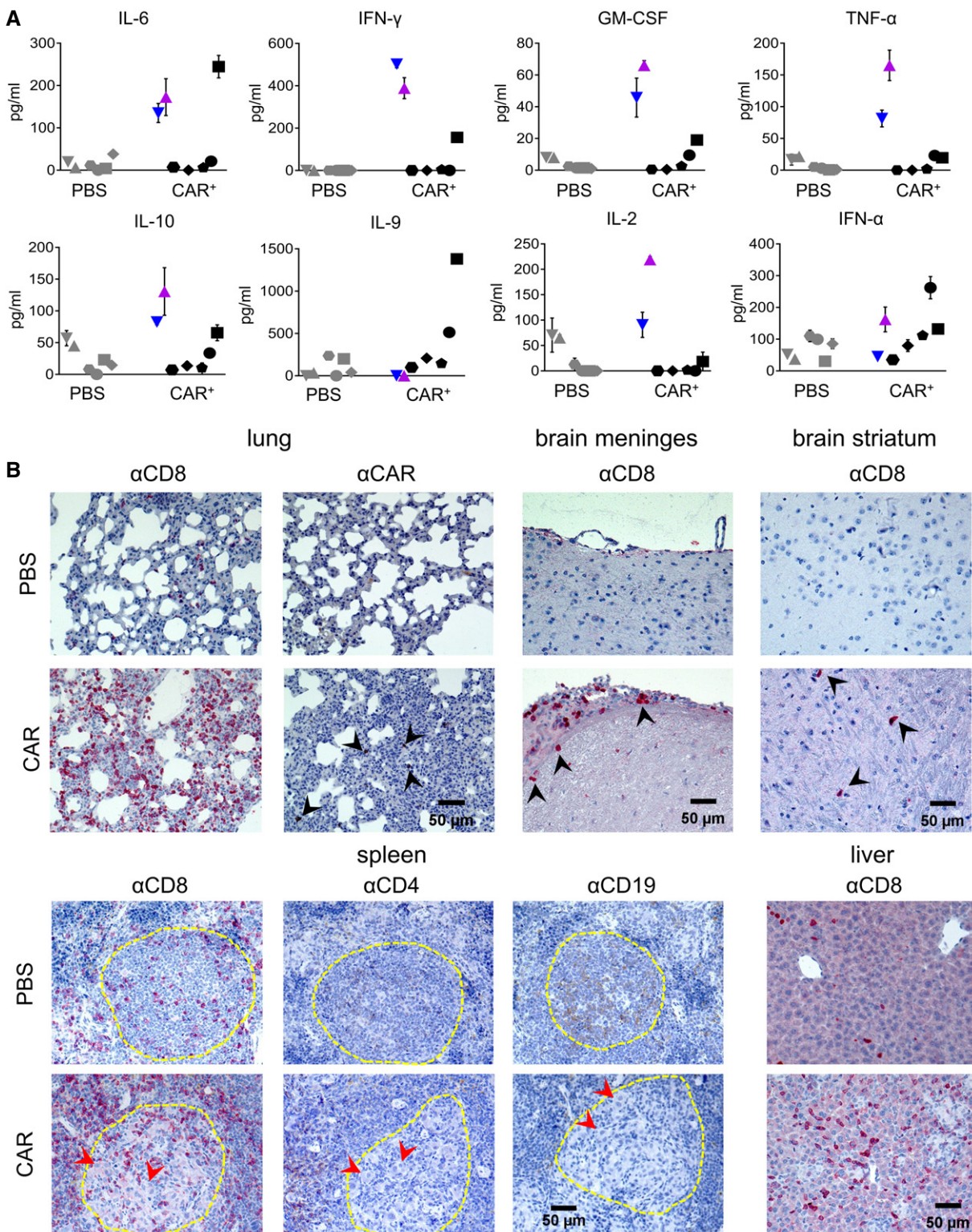

**Figure 3. Cytokines and histopathology.**

A    Cytokine levels in plasma of individual mice obtained from blood at 7 weeks after vector injection. The distinct symbols used for each individual mouse are identical to the ones used in Fig 2. Mean values ± SD of *n* = 2 technical replicates.

B    Immunohistochemistry of paraffin-embedded sections from the lungs, brain meninges, brain striatum, spleen, and liver of the vector-injected mouse M16 (CAR) and a control mouse injected with PBS (PBS) stained against CD8 (αCD8), the CAR (αCAR), CD4 (αCD4), or CD19 (αCD19). Black arrowheads point at infiltrated lymphocytes, red arrowheads point at histiocytes, and the yellow line indicates B-lymphocyte-rich zones.

    

detectable in lung, liver, and brain of M16 and M19 (Figs 3B and EV4). In contrast, spleen and bone marrow showed clear signs of lymphocytic depletion and acellular debris (Figs 3B and EV4). In the vector-treated mouse M16, the lungs displayed signs of interstitial pneumonia likely caused by many infiltrating CD8$^+$ T cells and also some CD4$^+$ T cells (Figs 3B and EV4). Some infiltrated lymphocytes were found to be myc-positive, although detection of the myc-tag incorporated in the CD19-CAR by immunohistochemistry is difficult due to poor sensitivity (Lobbestael *et al*, 2010; Fig 3B). Brain sections showed clear evidence for lymphocytic meningitis with CD8$^+$ and CD4$^+$ T lymphocytes mainly accumulating at the meninges, and some lymphocytes were also detected in the striatum (Fig 3B; Appendix Fig S5). Likewise, numerous infiltrations of CD8$^+$ lymphocytes were found in the liver (Fig 3B). The liver of M16 showed diffusely scattered aggregations of small epithelioid cell granulomas in the liver parenchyma and portal tracts consistent with a granulomatous hepatitis. The typical picture of hepatic GvHD with bile duct damage and endotheliitis was not observed (Fig EV5A). Likewise, colon tissue contained completely intact crypts with no sign of GvHD in ileum (Fig EV5B). In spleen, B-lymphocyte-rich zones reminiscent of primordial germinal centers were depleted of B cells and surrounded by many CD8$^+$ and some CD4$^+$ T cells (Fig 3B).

Taken together, these results provided proof of principle that CAR-encoding CD8-targeted LVs injected as "therapeutic drug" could induce development of human CD8$^+$ CAR T cells directly *in vivo*, where they conferred a potent cytotoxicity against CD19$^+$ cells. While this is the first demonstration for the *in vivo* generation of human CAR T cells, a recent publication described *in vivo* engineering of mouse T cells relying on the infusion of non-viral nanoparticles into mice (Smith *et al*, 2017). It is difficult to directly compare these two approaches, since the nanoparticles displayed an anti-CD3 targeting moiety, known to strongly activate and skew T-cell populations. Further, this new type of synthetic gene vector will likely require additional engineering and testing to exclude toxicity from the incorporated transposase plasmid before it can be applied for human T cells in patients. For the CD8-LV vector described here, translation into a clinical setting appears to be more straight-forward, since it transduces human T cells without requiring a strong T-cell activation signal and has been derived from LV vectors for which profound clinical experience is available (Naldini *et al*, 2016). Nevertheless, further preclinical testing will be necessary, including humanized mice combined with patient-derived tumors and experiments in large animals to establish the dose and pharmacology–toxicology of the genetic *in vivo* modification of CD8$^+$ T cells. Thereby, administration of the vector into lymph nodes may be an option to enhance the frequency of contacts between vector particles and target cells.

A surprising outcome of our study in fully humanized mice was the substantially increased concentrations of human cytokines in plasma of three of seven CAR-positive animals concomitantly with B-cell aplasia and the signs of neurotoxicity in one mouse. Human myeloid and CD4$^+$ T cells are the likely source for the released cytokines (Tanaka *et al*, 2012; Sundarasetty *et al*, 2017). The cytokine profile resembled that of CAR T-cell-treated patients suffering from CRS (Hay *et al*, 2017). However, in the clinical setting CRS appears much faster after CAR T-cell infusion, which is most likely due to the much higher numbers of *ex vivo*-activated CAR T cells

present immediately after infusion in the adoptive cell therapy approach. In contrast, after *in vivo* CAR gene delivery CAR T cells expand homeostatically and their levels rise more slowly. Preclinical testing of CAR T cells has been commonly performed in immunodeficient mice carrying human tumor xenografts. While such models provided evidence about the antitumor activity of the T cells, predictions about toxicities are difficult, because they are obscured by xenoreactivity and GvHD.

When *ex vivo*-generated CD19-CAR T cells were injected into HSC-transplanted mice, signs reflecting CRS similar to our study were observed, but could ultimately not be distinguished with absolute certainty from alloreactivities (Diaconu *et al*, 2017). In our setting, GvHD is highly unlikely. First, the CAR T cells are generated *in vivo* from T cells directly developing in the NSG mice from cord blood-derived CD34$^+$ cells, thus excluding alloreactivity. Second, GvHD does usually not develop in HSC-transplanted NSG. Only sporadic chronic GvHD with a late onset has been described in some mice, when transplanted with donor cells with an autoimmune-prone haplotype (Sonntag *et al*, 2015). Third, histology did not reveal any signs for GvHD when we assessed colon and the liver periportal tracts, which are typically affected by GvHD (Fig EV5). True CAR T-cell-induced CRS is therefore the most likely explanation for the observed symptoms, which is supported by a recent publication demonstrating CRS in HSC-transplanted mice upon injection with *ex vivo*-generated autologous CAR T cells (Norelli *et al*, 2018). Thus, it will be worth exploring our approach further toward its use as informative animal model allowing to study the mechanisms underlying CRS and neurotoxicity in CAR T-cell therapy (June *et al*, 2018). In this direction, it will be important to test the consequences of delivering the CAR not only into CD8$^+$ but also into CD4$^+$ T cells. While CD8$^+$ CAR T cells alone have been shown to be active against CD19$^+$ tumor cells *in vivo*, the simultaneous presence of CD4$^+$ CAR T cells boosts their activity (Sommermeyer *et al*, 2016). The tools for the simultaneous *in vivo* generation of CD4$^+$ CAR T cells are available (Zhou *et al*, 2015).

Successful implementation of *in vivo* CAR gene delivery in a clinical setting would have important consequences for the future development of adoptive immunotherapy, by shifting it from a highly personalized treatment to a broadly applicable off-the-shelf therapy: (i) The demanding and cost-intensive GMP-grade *ex vivo* manufacturing of T cells could be circumvented, (ii) vectors could be stored, shipped, and applied to patients when needed, and (iii) the increased flexibility would facilitate the development of novel immunotherapeutic concepts including indications beyond cancer and infectious disease.

## Materials and Methods

### Generation of vector particles and titration

The transfer vector plasmid pS-CD19-CAR-W coding for the CD19-CAR with the CD28ζ chain was generated by removing the IRES-GFP cassette from plasmid pS-63.28.z-IEW (Oelsner *et al*, 2016). The RFP-encoding transfer vector plasmid was generated by exchanging the luc-gfp cassette in pS-luc2-GFP-W (Abel *et al*, 2013) with the RFP reading frame. Stocks of CD8-LV were produced essentially as described (Bender *et al*, 2016). In brief, 0.9 μg of plasmid

pCAGGS-NiV-Gd34-CD8, 4.49 μg of plasmid NiV-F pCAGGS-NiV-Fd22, 14.5 μg of the packaging plasmid pCMVdR8.9., and 15.2 μg of the transfer vector encoding either CD19 CAR (pS-CD19-CAR-W) or RFP (pS-RFP-W) were cotransfected in $2.5 \times 10^7$ HEK-293T cells. After 2 days, vector particles released into the cell supernatant were concentrated and purified by ultracentrifugation through a 20% (wt/vol) sucrose cushion ($100,000 \times g$ for 3 h at 4°C or $4400 \times g$ for 24 h at 4°C). The supernatant was discarded, and pellets were resuspended in 50–60 μl of PBS. All vector stocks were frozen at −80°C until use.

To determine vector titers, MOLT 4.8 cells were transduced in serial dilutions and the percentage of CAR$^+$ cells was quantified by flow cytometry. Titer calculation was based on dilutions showing a linear correlation with the dilution factor. Average titers were calculated from duplicates.

### Cell culture

HEK293T cells were cultivated in DMEM (Gibco) supplemented with 10% fetal bovine serum (FBS; Pan Biotech). Routine tests for mycoplasma contamination were performed bi-annually. HEK293T cells were tested negative. Human PBMC were isolated from healthy anonymous donors that had given informed consent, or from buffy coats purchased from the German Red Cross blood donation center (DRK-Blutspendedienst Baden-Württemberg-Hessen, Frankfurt). PBMC were cultivated in RPMI 1640 medium supplemented with 10% FBS, 2 mM L-glutamine, 0.5% penicillin/streptomycin, 25 mM HEPES (Sigma-Aldrich), and 100 IU/ml IL-2 (Miltenyi Biotec). For activation, PBMC were seeded on plates precoated with 1 μg/ml anti-human CD3 mAb (clone OKT3, Miltenyi Biotec) and 3 μg/ml anti-human CD28 mAb (clone: 15E8, Miltenyi Biotec) was added to the cell culture medium for 72 h.

### CAR T-cell-mediated cell killing

The cytotoxicity of T cells was determined by a flow cytometry-based assay. CAR expression was analyzed prior to the killing assay in order to compensate for untransduced cells. Only CAR$^+$ T cells were counted as effectors. Target cells were labeled with CFSE (Thermo Fisher Scientific). CAR T cells were added to $5 \times 10^4$ target cells in round-bottom 96-well plates in ratios ranging from 5:1 to 0.15:1 and co-cultivated for 4 h at 37°C. Dead cells were identified by staining with the fixable viability dye eFluor450 (Affymetrix Bioscience). The percent killing of target cells was calculated by multiplying the ratio of the number of dead CFSE$^+$ cells and the total cell number of CFSE$^+$ cells with 100.

### Gene transfer

For transduction of primary human PBMC, $1 \times 10^5$ cells were seeded per well of a 48-well plate. Vector was added to the cells, and spinfection was performed by centrifugation at $850 \times g$ for 90 min at 32°C. Percentages of CAR$^+$ T cells were determined by flow cytometry 5 days post-transduction.

For *in vivo* gene transfer, animal experiments were performed in accordance with the regulations of the German animal protection law and the respective European Union guidelines. NSG mice (NOD.Cg.Prkdc$^{scid}$IL2rg$^{tmWjl}$/SzJ, Jackson Laboratory) were i.p.

injected with $1 \times 10^7$ activated human PBMC, followed by i.p. injection of $2 \times 10^6$ t.u. of CD8-LV encoding either CD19-CAR or RFP. Tumor-bearing mice were generated by injection with $5 \times 10^5$ Raji cells 6 days before PBMC engraftment. A few NSG mice (< 5), which had been engrafted with human PBMC, but did not show human engraftment (determined via flow cytometry analysis using the human CD45 antibody), were excluded. Seven days after vector application, blood was taken, mice were sacrificed, and peritoneal lavage was performed to obtain peritoneal cells. Spleen and lungs were removed, and single cell suspensions from spleen were obtained by meshing the tissue through a 45-μm cell strainer. Lung single cell suspensions were generated according to the protocol of the lung dissociation kit (Miltenyi Biotec). Single cell suspensions from blood, spleen, and peritoneum were analyzed by flow cytometry.

Human CD34$^+$ cord blood-engrafted mice were purchased from Jackson Laboratory as hu-CD34-NSG™ (transplanted upon irradiation with human CD34$^+$ cord blood cells at 3 weeks of age). Mice were checked for the amounts of human CD45$^+$ cells in blood via FACS analysis prior to the start of the experiment and randomly allocated to the vector and PBS control groups. Investigators were not blinded to group allocation. No mice were excluded. Two times 200 ng human IL-7 was injected intravenously into all hu-NSG mice 5 and 1 day before intravenous injection of $2 \times 10^6$ t.u. of CD8-LV$^{CD19CAR}$ or PBS. The presence of CAR T cells in blood was monitored by FACS analysis using an antibody against the myc-tag incorporated in the CAR. Once animals were sacrificed, cells from blood, spleen, and bone marrow were harvested for flow cytometric analysis. Further tissues were fixed in 4% formalin and embedded in paraffin for immunohistochemistry.

### Histology

Paraffin blocks were cut into 4-μm-thick sections, mounted on silanized glass slides, and subsequently deparaffinized and rehydrated using xylene and graded ethanol washes, followed by washing with phosphate-buffered saline (pH 7.4). Slides were directly stained with hematoxylin/eosin or used for immunohistochemistry. Immunohistochemistry was performed after heat-induced epitope retrieval at pH 8 in a pressure cooker. Primary antibodies for detection of CD8 (monoclonal mouse anti-human CD8, clone C8/144B, DAKO, Glostrup, Denmark, 1:100), CD4 (monoclonal mouse anti-human CD4, clone 4B12, Ready-to-use, DAKO), and CD19 (monoclonal mouse anti-human CD19, clone LE-CD19, Ready-to-use, DAKO) or for the myc-tag of the CAR construct (9E10, Thermo Fisher Scientific) were incubated for 30 min or 2 h (myc-tag), respectively. For detection, the FLEX-Envision (DAB) or K5005 (Permanent Red) Kit (DAKO) were applied according to the manufacturer's instructions. Histological sections were blinded before evaluation by an expert pathologist.

### Cytokine profiling

Plasma was harvested from the peripheral blood of mice by centrifugation for 5 min at $300 \times g$ at room temperature. Supernatant was transferred into fresh tubes and centrifuged again for 5 min at $2,000 \times g$. Plasma was stored at −80°C until analysis with MACS-Plex Human Cytokine 12 Kit (Miltenyi Biotec) following the

manufacturer's instructions. Samples were analyzed using the MACSQuant Analyzer 10 and the MACSQuantify™ 2.8 software (Miltenyi Biotec).

## Flow cytometry

Flow cytometric analysis was performed on the LSRII (BD Biosciences) or MACSQuant Analyzer 10 (Miltenyi Biotec), and data were analyzed using FCS Express V6 (De Novo Software) or FlowJo V10.1 (FlowJo). Human T cells were identified with antibodies directed to hCD45 (clone: 5B1), hCD3 (clone: BW264/56), and hCD8 (clone: BW135/80; all Miltenyi Biotec). The gating schemes for the PBMC-transplanted (Appendix Fig S1) and the CD34-transplanted mice (Appendix Fig S4) are provided. The percentage of human CD45$^+$ cells in the peripheral blood of HSC-transplanted mice was calculated as the amount of human CD45$^+$ cells within all single, viable, and mononucleated cells. CD19 expression was analyzed with an anti-human CD19 antibody (clone: LT19, Miltenyi Biotec). CAR cell surface expression was detected via the myc-tag incorporated in the CAR using a PE-labeled anti-myc-tag antibody (clone: 9B11, Cell Signaling Technology). To evaluate T-cell exhaustion, antibodies directed against PD-1 (clone PD1.3.1.3), LAG-3 (clone REA351), and TIM-3 (clone REA635, all from Miltenyi Biotec) were applied. For all *in vivo* analyses, cells were incubated with mouse Fc block (Miltenyi Biotec). Dead cells were excluded from the analysis using LIVE/DEAD™ Fixable Dead Cell Stain (Thermo Fisher Scientific).

## Quantification of vector copy numbers and LM-PCR

Vector copy number (VCN) analysis of mice engrafted with human PBMC was performed by TaqMan-based qPCR using a LightCycler® 480 Instrument II (Roche), and data were analyzed with Light-Cycler® Software. VCN were determined on genomic DNA, isolated from the indicated tissues. Transgene detection was performed using the woodchuck hepatitis posttranscriptional element (WPRE)-specific probe (5′-tgcactgtgtttgctgacgcaac-3′) and primers (fwd: 5′-tcctggttgctgtctctttatg-3′ and rev: 5′-tgacaggtggtggcaatg-3′). As an internal reference, a human albumin-specific probe (5′-acgtgaggag tatttcattactgcatgtgt-3′) and primers (fwd: 5′-cacactttctgagaaggagagac -3′ and rev: 5′-gcttgaattgacagttcttgctat-3′) were used. A plasmid standard containing sequences of WPRE and human albumin was used for quantification. VCN were calculated as the ratio of (copies WPRE)/(copies albumin). VCNs in vector-injected HSC-engrafted NSG mice were determined on genomic DNA isolated from bone marrow-derived enriched CD8$^+$ T cells. CD8$^+$ enrichment was performed using 3 μl of APC-labeled anti-CD8 antibody (Miltenyi Biotec) following the MACS anti-APC bead enrichment with MS columns according to the manufacturer's instructions. Average vector copy numbers were determined by qPCR with primers amplifying the packaging signal (primers: fwd: 5′-tgtgtgcccgtctgttgtgt-3′; rev: 5′-gagtcctgcgtcgagagagc-3′; probe: 5′-cagtggcgcccgaacaggga-3′) after normalization for endogenous beta-actin genes (primers: fwd: 5′-tccgtgtggatcggcggctcc-3′; rev:5′-ctgcttgctgatccacatctg-3′; probe: 5′-cctggcctcgctgtccaccttcca-3′). Results were compared with those obtained after serial dilutions of genomic DNA from a cell line containing one copy of the integrated lentiviral vector per haploid genome.

## The paper explained

### Problem

Immunotherapy has brought impressive benefit to cancer patients not only in clinical trials but also with an increasing number of products on the market. Among these are genetically engineered chimeric antigen receptor (CAR) T cells. CARs are artificially designed receptors composed of a tumor antigen-specific binding domain and an intracellular signaling domain, which transfers activation signals upon tumor antigen binding. Thus, T cells equipped with CD19-specific CARs become activated when recognizing lymphoma cells, proliferate, and kill the tumor cells. CAR T cells are regarded as an individualized medicinal product, since they are produced from the patients' autologous T cells which will be genetically modified *ex vivo*, using lenti- or retroviral vectors, then be expanded and finally re-infused back into the patient. This process means not only high production costs but also submitting the T cells to various modifications thereby altering their phenotypes and activities.

### Results

Here, we provide evidence that CAR T cells can be generated directly *in vivo* by injection of lentiviral vectors delivering the CD19-CAR precisely into CD8$^+$ T cells. A single injection of vector particles into mice transplanted with human blood cells was sufficient to detect CAR T cells in blood and lymphoid organs. These *in vivo*-generated CAR T cells expanded upon antigen recognition and eliminated CD19-positive cells. Evidence for *in vivo* CAR T-cell generation was also collected in a mouse model containing resting human lymphocytes suggesting that the clinical translation of this strategy is possible. Interestingly, some of the mice developed a cytokine release syndrome with infiltrating lymphocytes in spleen, liver, and brain resembling what has been observed in some patients that were treated with CAR T cells.

### Impact

This is the first demonstration that human cytotoxic T cells can be selectively genetically modified directly *in vivo* and are reprogrammed to recognize and kill target cells. This proof-of-concept study has far-reaching implications. Once this approach has been translated into clinics, CAR T-cell therapy can become much simpler and more easily accessible to patients. Beyond that, our approach may open novel options for the *in vivo* manipulation of T cells in immunotherapy and immunology in general.

Integration site analysis of isolated PBMC from peritoneal cavity or spleen of NSG mice was performed by ligation-mediated PCR (LM-PCR) amplifying the 3′ long terminal repeat (LTR) junction as described before (Friedel *et al*, 2016). Briefly, the DNeasy Blood and Tissue Kit (Qiagen) was used to purify genomic DNA, which was digested with Tsp509I (Thermo Fisher Scientific) and then precipitated with 100% EtOH, 20 μg glycogen, and 0.1 M Na-acetate. Primer extension was performed with 0.5 pmol of biotinylated primer lvLTR1 (bio-5′-gaacccactgcttaagcctca-3′). Biotinylated, newly synthesized strands were purified using the QIA Quick PCR Purification Kit (Qiagen) and enriched by magnetic bead capture. Captured magnetic beads were washed twice with 100 ml water, resuspended in ligation mix containing the linker oligos and 80 U T4-Ligase (New England Biolabs), and incubated at 16°C overnight. The first PCR was performed with the primers lvLTR2 (5′-agcttgcctt gagtgcttca-3′) and OCI (5′-gacccgggagatctgaattcg-3′), the following nested PCR with primers lvLTR3 (5′-agtagtgtgtgcccgtctgt-3′) and OCII (5′-agtggcacagcagttaggacg-3′).

## Statistical analysis

This was an explorative study with unknown expectations about its outcome. Therefore, sample sizes were estimated to observe statistical differences. For a certain sample size, we estimated which differences can be statistically detected. To observe differences in vector-treated and PBS control mice, we set transduced cells as factor observed. By using single-factor variance analyses for sample sizes of $n \geq 3$ (significance level = 0.05, power = 80%), differences to the mean by 1.84-fold of the standard deviation can be observed as statistically relevant.

All individual animals were randomized by chance. All data are displayed as mean and standard deviation (SD). Statistical significance between two groups was determined using unpaired Student's *t* and Mann–Whitney tests. Statistical significance between three or more groups was calculated using one-way ANOVA with Bonferroni correction. Only data positive for normal distribution by Shapiro–Wilk test were assessed. *P*-values are given in the figure legends. GraphPad Prism 5 software was used for statistical analysis.

Expanded View for this article is available online.

## Acknowledgements

The authors wish to thank Jörg Kirberg for helpful discussions and Gundula Braun as well as Tatjana Weidner (PEI) for producing vector particles. This work was supported by grants from the Deutsche Krebshilfe (70112587) to C.J.B. and the LOEWE Center for Cell and Gene Therapy Frankfurt funded by Hessisches Ministerium für Wissenschaft und Kunst (IIL5-518/17.004) to I.C.S. and C.J.B.

## Author contributions

AP and FBT planned and performed the experiments, analyzed data, and contributed to writing of the manuscript. SH performed and evaluated histology. AMF performed and evaluated exhaustion marker staining. SD generated HSC-transplanted mice and advised with flow cytometry settings. CC determined vector copy numbers. RRB, WSW, UM, RS, and EV provided unique reagents and protocols and advised on their usage. RS, WSW, and EV contributed to writing of the manuscript. CJB initiated and supervised the project, acquired grants, planned experiments, and wrote the manuscript.

## Conflict of interest

R.B., E.V., and C.J.B are listed as inventors on a patent covering the CD8-targeted lentiviral vector. All other authors declare that they have no conflict of interest.

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
