## [Review Process File · EMBO Molecular Medicine]

***In vivo* generation of human CD19-CAR T cells results in B cell depletion and signs of cytokine release syndrome**

Anett Pfeiffer, Frederic B. Thalheimer, Sylvia Hartmann, Annika M. Frank, Ruben R. Bender, Simon Danisch, Caroline Costa, Winfried S. Wels, Ute Modlich, Renata Stripecke, Els Verhoeyen, Christian J. Buchholz.

Review timeline:

Submission date:	25 th March 2018
Editorial Decision:	3 rd May 2018
Additional correspondence:	11 th May 2018
Additional correspondence:	14 th May 2018
Revision received:	18 th June 2018
Editorial Decision:	6 th August 2018
Revision received:	14 th August 2018
Accepted:	17 th August 2018

Editor: Lise Roth

Transaction Report:

1st Editorial Decision

3rd May 2018

Thank you for the submission of your manuscript to EMBO Molecular Medicine. We have now received feedback from two of the three reviewers who agreed to evaluate your manuscript. Given that we have not heard from referee 3 despite several chasers and that both referees 1 and 2 are overall positive, we prefer to make a decision now in order to avoid further delay in the process. Should we receive the report from referee 3 within 1 to 2 weeks from now, we will send it to you. In that event, please note however that you will only be asked to address this referee's minor concerns (nothing further reaching).

***** Reviewer's comments *****

Referee #1 (Remarks for Author):

This manuscript describes a strategy to generate human CD19-CAR T cells *in vivo*, and to thereby target CD19+ B cells. Two humanized mouse models are used: (1) engrafted with human PBMC + the Raji tumor cell line, and (2) engrafted with human HSC. FACS analysis and qPCR for vector genomes are used to evaluate CD19-CAR expression in human CD8+ cells in different tissues, and B cell depletion is examined by FACS staining for CD19+ cells. At the time of sacrifice, both PBMC and HSC mice showed lower levels of CD19+ cells in animals injected with CD8-targeted lentiviral vectors expressing the CD19 CAR (CD8-LVCD19CAR) when compared to PBS control mice. Further analysis of cytokine profiles and histology suggested signs of cytokine release syndrome in CD8-LVCD19CAR injected mice.

Overall, these findings are potentially interesting and relevant to the development of an *in vivo* gene therapy approach to generate CAR T cells. However, I have several concerns about the data which are described below:

Major concerns

1. For the PBMC and Raji tumor cell mouse model:

(a) Convincing evidence of B cell depletion requires measurements of the B cell levels in the blood of individual

mice before CD8-LVCD19CAR administration, not simply a comparison to control mice.

(b) It is not clear whether Raji cells have contributed to B cell engraftment in this model. The overall B cell levels at the time of sacrifice are very low (average value <1% in all condition). In addition, B cell frequencies were similar in CD8-LVRFP injected mice whether Rajis were injected or not. This was also observed in the frequency of CD8-LVCD19-CAR injected mice group (Fig. 1F).

(c) Please provide more detailed gating schemes for the FACS used in these mice, including CD45 percentages and the strategy used to assess human cell expansion and for downstream analysis

(d) The vector copy number detected by DNA analysis does not appear to align with analysis by flow cytometry. Copy numbers seem similar to the frequency of CAR+ CD8+ T cells reported in panel E (assuming 1 copy per cell). However, the text indicates that genomic DNA was isolated from total tissue. While murine cells should be irrelevant assuming the control PCR is human-specific, the CD8+ fraction represents half or less of the CD3+ cells in all tissues and an unknown percentage of the total human cells. Thus, this VCN seems at least 2x as high as would be expected from the flow cytometry data. (Fig. 1C, 1D). Please comment.

(e) For detecting control transduced T cells, is it fair to compare RFP expression to a myc-tag antibody? Are these reporters equally sensitive and do both correlate well with VCN during, for example, in vitro transduction analysis such that this is a reasonable comparison?

(f) Methods do not detail how PBMCs were activated prior to injection.

(g) Figure 1D shows higher CD8+ cell frequency in CD8-LVCD19CAR injected mice than in CD8-LVREF injected mice, and the authors speculate that this is due to expansion of CD8+CD19CAR+ cells upon antigen stimulation by B cells. However, this higher expression was only observed in the peritoneum, and the reason for the differences between tissue is not discussed. Moreover, in Figure 1E, the higher level of transgene expression in CD8-LVCD19CAR injected mice than in CD8-LVREF injected mice was explained as being due to CD8+ cell expansion. However, since the "expansion" was only observed in the peritoneum, it is unclear why the transgene level in spleen and blood also has this difference between CD8-LVCD19CAR and CD8-LVREF injected mice.

(h) If the T cell expansion is indeed driven by CAR mediated stimulation, it would be interesting to know how diverse the T cell repertoire of the CAR+ cells is. Do they come from massive expansion of a single cell, or are they polyclonal? Could this have implications for the efficacy of these responses i.e. could the cells become more easily exhausted and nonfunctional if the starting population was not sufficiently robust?

2. The efficacy of the CAR T cells is only evaluated against injected Raji cells into animals. It would be informative to see whether this therapy is able to protect animals or eliminate a tumor in a xenograft model.

3. Questions about the CD34-NSG mouse experiments:

(a) The vector copy number detected by DNA analysis does not appear to align with analysis by flow cytometry. The qPCR detected vector copy number was around 1 copies/human genome in bone marrow tissue enriched for CD8+ cells, while the FACS analysis in panel D showed only 5% of the cells expressing CAR (and it's not clear this 5% is under human CD45+ gate or total lymphocyte gate, the information about the gating method is poorly provided).

(b) The interpretation that the pathology observed in mice M16 and M19 was related to CRS seems a little premature. While this is a reasonable hypothesis, GvHD is not unheard of in CD34-NSG mice, particularly in animals with high levels of circulating human CD45+ cells in the blood, such as the up to 72% reported in the text. What were the humanization levels in these animals and how did they compare to levels in the PBS control group? The reported random assignment might have resulted in erroneous segregation of highly-humanized animals to one group, rather than using a rank-ordered system to ensure equivalent starting means among the groups.

(c) Alternatively, the authors could better support their CRS hypothesis if they presented the initial humanization and lymphocyte subset data for the animals and could link CRS to B cell "burden." If M16 and M19 had the highest initial levels of B cells, that might suggest antigen burden was a risk factor for this pathology, as has been observed in clinical trials of CAR T cells.

(d) CD34-NSG mice don't have germinal centers. The structures outlined as such in Fig. 3 do not resemble a germinal center, lacking the appropriate density of nuclei and B cells and with no indications of light/dark zones. While clusters slightly enriched for B cell density can be observed in this model, I am not aware of any literature suggesting that these possess the most critical hallmarks of germinal centers. Moreover, the fact that different stains were done on images that are not serial sections further challenges any interpretation of this data.

(e) In Figures 2E-F, it appears that animals negative for CAR expression by flow and VCN (open symbols) were excluded from analyses of B cell depletion. This cherry-picking of the data is not justified, and it excludes a potentially interesting finding if B cell depletion was also observed in these animals.

(f) Why is CAR expression so much dimmer in the CD34 model than the PBMC mice? There is a sentence in the text suggesting this is expected, but no reference is cited to support this interpretation. Is this a known characteristic of the SFFV promoter (which is suggested to be used by the reference but not indicated in this manuscript)?

(g) The manuscript states that HSC-mice were used "to assess if CAR T cells could also be generated from T cells in steady-state....". However, IL-7 was used in order to activate CD8+ T cells in the HSC-mice prior to vector injection. Evidence/appropriate controls are needed to show that the IL-7 injections induce proliferation of T cells or increase transduction in vivo.

Minor concerns

- Typo on 6th line of the results and discussion: "asses"
- Fig EV 2 looks like its missing some labels. Why are there 3 repeats of PBS/CAR/RFP? Are these the three tissues in order? If so, the data doesn't appear to align with Fig 1G as indicated.
- Why do only some histology images have scalar bars? At least 1 should be provided per each tissue.
- Figure 1B and 2A need more detail on the graph. Ex: what type of vector is injected and what are the does were being used. These were explained separately in the legend or method but it is hard for the reader to gather all the information.
- For figure 1D-G, a represented FACS gating was presented. The tissue of the FACS gating should be labeled on the plot.

Referee #2 (Comments on Novelty/Model System for Author):

The paper by Pfeffer et al on in vivo generation of CD19 CAR-T cells is of major translational importance for medicine and for your journal. In some ways it is a breakthrough in the field of CAR-T. First was Zelig Eshhar's concept, but while he could technically make them, they were poorly activated in the absence of the appropriate signalling entities. Second was led by June et al adding signalling sequences to CAR which led to fully functional ex-vivo generated CD19 CAR-T, clinically effective.

But at a cost of \$500K per patient! While its not this journals reviewers role to consider the health economics, it is clear that CAR-T are not scalable to all due to problems of manufacture as well as cost. Bucholtz' group here report the first clear steps towards a much simpler scalable and hence cheaper approach generating CAR-T in vivo. This is succinctly and effectively reported. Not only is there efficacy but all the variations and side effects are also present in mice receiving vectors to make CART suggesting what they made is very similar to existing products generated exvivo. The work is well documented and they are very modest about the implications of their work.

Additional correspondence – Author's comment

11th May 2018

We were pleased seeing that the two reviewers were both overall very positive about our manuscript. We have gone through the points raised by reviewer 1 and will be able to address almost all of them providing additional data and/or better explanation/discussion in the text. An exception is point #2 in which this reviewer asks for a xenograft tumor model. This is unfortunately not so easy to set up for the in vivo CAR delivery and will take some time to get this done. One reason is that we have to deal with a strong alloreactivity between the transplanted human PBMC and the tumor cells. In conventional CAR T cell experiments this is less of an issue since a high dose of CAR T cells is injected into the animals whereas in our setting CAR T cells develop only slowly within a large surplus of non-CAR PBMC. Given in addition that the CAR T cell field is moving fast we feel that this point goes beyond the scope of the current manuscript where we provide in a report format proof-of-principle for the in vivo CAR T generation and describe CRS-like side-effects in the mice. I'd therefore be grateful if you could give us some indication how essential this point 2 by reviewer 1 will be seen upon revision of the manuscript.

Additional correspondence - Referee #1's comment

14th May 2018

[The author] makes a reasonable point, and I would be OK to accept this argument.

1st Revision - authors' response

18th June 2018

***** Reviewer's comments *****

Referee #1 (Remarks for Author):

This manuscript describes a strategy to generate human CD19-CAR T cells in vivo, and to thereby target CD19+ B cells. Two humanized mouse models are used: (1) engrafted with human PBMC + the Raji tumor cell line, and (2) engrafted with human HSC. FACS analysis and qPCR for vector genomes are used to evaluate CD19-CAR expression in human CD8+ cells in different tissues, and B cell depletion is examined by FACS staining for CD19+ cells. At the time of sacrifice, both PBMC and HSC mice showed lower levels of CD19+ cells in animals injected with CD8-targeted lentiviral vectors expressing the CD19 CAR (CD8-LVCD19CAR) when compared to PBS control mice. Further analysis of cytokine profiles and histology suggested signs of cytokine release syndrome in CD8-LVCD19CAR injected mice.

Overall, these findings are potentially interesting and relevant to the development of an in vivo gene therapy approach to generate CAR T cells. However, I have several concerns about the data which are described below:

Major concerns

1. For the PBMC and Raji tumor cell mouse model:

(a) Convincing evidence of B cell depletion requires measurements of the B cell levels in the blood of individual mice before CD8-LVCD19CAR administration, not simply a comparison to control mice.

We now provide these data for the CD34-NSG model (Appendix Fig. S4B). There is no difference in the CD19 levels between the vector injected group and the control group. Moreover, Fig. 2F shows the levels of CD19+ cells after treatment related to the levels before treatment. There is a statistically significant reduction of the B cell levels in the CAR+ group only.

In the PBMC-transplanted mice such an analysis was not compatible with our experimental setting, since we administered PBMC into the peritoneal cavity and 24 hours later the vector particles. Since migration of PBMC from the peritoneal cavity to other organs can take 7-14 days (King et al, 2008, Clin. Immunol. 126:303-314) there were no B cells to be expected in blood at this early time point. Moreover, when we compile all data collected in this mouse model and compare the B cell levels in CAR- versus RFP/PBS-injected mice, the statistical probability for the B cell depletion being accidental is below 0.0001. Please see the figure below for your information.

Moreover, we have now added a data set demonstrating that cells transduced with CD8-LV delivering the CD19-CAR also deplete B cells from human PBMC in vitro (Fig. EV1B).

Summary of the data for CD19⁺ cell elimination in PBMC-transplanted mice.

(A) The transgene level of CD8⁺ cells from peritoneal cells is shown for mice treated with CD8-LV(CAR) (filled circle) and for mice treated with PBS or CD8-LV(RFP) (open circle) (B) human CD19 levels within human CD45⁺ cells harvested from the peritoneal cavity. Mean values ± SD is shown with n=15 (CAR group) and n=17 (control group). Statistical significance was determined by two-tailed unpaired t-test, **** p<0.0001.

(b) It is not clear whether Raji cells have contributed to B cell engraftment in this model. The overall B cell levels at the time of sacrifice are very low (average value <1% in all condition). In addition, B cell frequencies were similar in CD8-LVRFP injected mice whether Rajis were injected or not. This was also observed in the frequency of CD8-LVCD19-CAR injected mice group (Fig. 1F).

The CD19+ cells in blood and peritoneal cavity were mainly B lymphocytes whereas Raji cells had rather invaded into tissues, especially the peritoneum, by the time of harvest. This was confirmed by using CD45 expression levels to distinguish between B cells (higher expression) and Raji cells (lower expression) (Appendix Fig S3). We do now explicitly state this in the manuscript on page 6, 1st paragraph.

(c) Please provide more detailed gating schemes for the FACS used in these mice, including CD45 percentages and the strategy used to assess human cell expansion and for downstream analysis

The gating scheme is now provided in the Appendix Fig. S1A, the percentages of CD45+ and CD3+ cells in Appendix Fig. S1B-C. This analysis shows that there is no expansion of human cells in general. There is, however, a relative increase of the CAR+ T cells which we demonstrated by comparing to RFP gene delivery and by B cell depletion. We do now mention this on page 5.

(d) The vector copy number detected by DNA analysis does not appear to align with analysis by flow cytometry. Copy numbers seem similar to the frequency of CAR+ CD8+ T cells reported in panel E (assuming 1 copy per cell). However, the text indicates that genomic DNA was isolated from total tissue. While murine cells should be irrelevant assuming the control PCR is human-specific, the CD8+ fraction represents half or less of the CD3+ cells in all tissues and an unknown percentage of the total human cells. Thus, this VCN seems at least 2x as high as would be expected from the flow cytometry data. (Fig. 1C, 1D). Please comment.

To facilitate comparison between VCNs and FACS numbers we provided the percentages of CAR+ cells among human CD45+ cells in Fig. EV2. We realized that the bars were wrongly labelled for the CAR and RFP groups. Having now corrected this labelling error, we believe that the FACS and VCN data fit perfectly well (please compare Fig. EV2 and Fig 1C), at least in their relative numbers. Overall, the VCNs may still be a little higher than the expression levels determined by FACS which can be due to multiple integrations per cell or loss of gene expression in transduced cells. It is also well established that the correlation between the number of transduced cells and the VCN is linear only up to about 30% gene transfer. Beyond that, the VCNs increase much faster (Kustikova et al., 2003; Blood 102, 3934-3937; Fehse et al., 2004; Gene Therapy 11, 879-881).

(e) For detecting control transduced T cells, is it fair to compare RFP expression to a myc-tag antibody? Are these reporters equally sensitive and do both correlate well with VCN during, for example, in vitro transduction analysis such that this is a reasonable comparison?

We have for both detection systems a proper signal to noise ratio and are therefore convinced that we can clearly identify CAR+ as well as RFP+ cells by flow cytometry. This is supported by the data detecting transduced cells on the genomic level by determining VCNs. As explained above these correlate very well for the different groups assessed.

(f) Methods do not detail how PBMCs were activated prior to injection

PBMC were activated via CD3/CD28 prior to injection into NSG mice. The activation protocol was provided in the Methods section (see page 13, paragraph labelled "Cell Culture" in the revised manuscript).

(g) Figure 1D shows higher CD8+ cell frequency in CD8-LVCD19CAR injected mice than in CD8-LVREF injected mice, and the authors speculate that this is due to expansion of CD8+CD19CAR+ cells upon antigen stimulation by B cells. However, this higher expression was only observed in the peritoneum, and the reason for the differences between tissue is not discussed. Moreover, in Figure 1 E, the higher level of transgene expression in CD8-LVCD19CAR injected mice than in CD8-LVREF injected mice was explained as being due to CD8+ cell expansion. However, since the "expansion" was only observed in the peritoneum, it is unclear why the transgene level in spleen and blood also has this difference between CD8-LVCD19CAR and CD8-LVREF injected mice.

We agree that the enhanced CD8⁺ level was especially pronounced in the peritoneal cavity (we are now stating this in the manuscript on page 5), however a tendency for an increase is also detectable in spleen and blood. In any case, the CD8 expansion in the peritoneal cavity did not directly correlate with the CAR expression levels. The enrichment for CD8 cells is below 2-fold compared to the control groups, but the enrichment for CAR⁺ cells is at least 10-fold on the protein level (FACS) and 4-5-fold on the genomic level (VCN). We state in the manuscript that this must be due to a preferential expansion of the transduced cells, i.e. the CAR⁺ cells (not the CD8⁺ cells overall). Levels in spleen and blood require extravasation of CAR⁺ T cells from the peritoneal cavity. Our clonality analysis (see below) revealed that not all but only a fraction of CAR⁺ T cells migrated from the peritoneal cavity to blood and spleen. This is well in agreement with the CD8⁺ and CAR⁺ levels we detected in these compartments. We are now explaining this in the manuscript on page 6, 2nd paragraph.

(h) If the T cell expansion is indeed driven by CAR mediated stimulation, it would be interesting to know how diverse the T cell repertoire of the CAR⁺ cells is. Do they come from massive expansion of a single cell, or are they polyclonal? Could this have implications for the efficacy of these responses i.e. could the cells become more easily exhausted and nonfunctional if the starting population was not sufficiently robust?

We now provide data on the clonality of the in vivo generated CAR T cells generated by PCR on genomic DNA specific for the integration site of the vector. The data show that we have a clear polyclonal situation in the peritoneum in the presence of B cells (Fig. EV3A). This is again in agreement with a preferential expansion of CAR⁺ cells. In absence of B cells and for RFP gene transfer the transduced cells are more oligoclonal, since distinct bands can be detected in the integration site PCR (LM-PCR). In spleen, we have a clear oligoclonal pattern suggesting that only distinct CAR T cell subpopulations were able to migrate from the peritoneum to spleen.

As suggested, we have measured the activation/exhaustion markers PD1, LAG3 and TIM3 in spleen and peritoneal cavity cells from mice injected with CAR vector, RFP vector or PBS, respectively (Fig. EV3B). The data reveal a significantly increased level of exhaustion in a fraction of the CAR⁺ cells.

We now describe this on page 6, 2nd paragraph.

2. The efficacy of the CAR T cells is only evaluated against injected Raji cells into animals. It would be informative to see whether this therapy is able to protect animals or eliminate a tumor in a xenograft model.

We agree that this is a logical next step to evaluate. It is, however, not so easy to set up for the in vivo CAR delivery and it will take some time to get this done. One reason is that we have to deal with a strong alloreactivity between the transplanted human PBMC and the tumor cells. In conventional CAR T cell experiments this is less of an issue since a high dose of CAR T cells is injected into the animals whereas in our setting CAR T cells develop only slowly within a large surplus of non-CAR PBMC. Given in addition that the CAR T cell field is moving fast we feel that this point goes beyond the scope of the current manuscript where we provide proof-of-principle for the in vivo CAR T generation and describe CRS-like side-effects in the mice.

3. Questions about the CD34-NSG mouse experiments:

(a) The vector copy number detected by DNA analysis does not appear to align with analysis by flow cytometry. The qPCR detected vector copy number was around 1 copies/human genome in bone marrow tissue enriched for CD8⁺ cells, while the FACS analysis in panel D showed only 5% of the cells expressing CAR (and it's not clear this 5% is under human CD45⁺ gate or total lymphocyte gate, the information about the gating method is poorly provided).

We now provide more details on the gating strategy (see Appendix Fig. S4). Human CD8⁺ cells were gated as % of on hCD3⁺ cells in the viable human CD45⁺ cells from bone marrow cells. The qPCR was mainly performed to distinguish between CAR⁺ and CAR⁻ mice. We agree that although overall the FACS and VCN data fit well to each other, the VCNs suggest on average more CAR⁺ cells than determined by FACS. Multiple integrations in single cells as well as inactivation of the SFFV promoter used in our constructs, which has been previously observed (Stein et al., 2010; Nat Med 16, 198-204), are possible explanations. The IL-7 stimulation is only transient with cells

returning to a resting state a few days later (see below). Such resting or minimally activated T cells may not express the CAR in sufficiently high levels to be detected by flow cytometry, especially when carrying only a single integration. We now explain this on page 7, 1st paragraph.

(b) The interpretation that the pathology observed in mice M16 and M19 was related to CRS seems a little premature. While this is a reasonable hypothesis, GvHD is not unheard of in CD34-NSG mice, particularly in animals with high levels of circulating human CD45+ cells in the blood, such as the up to 72% reported in the text. What were the humanization levels in these animals and how did they compare to levels in the PBS control group? The reported random assignment might have resulted in erroneous segregation of highly-humanized animals to one group, rather than using a rank-ordered system to ensure equivalent starting means among the groups.

We now provide the humanization levels in both mouse groups including CD45+, CD19+, CD3+ and CD8+ cells. There is random allocation of mice to the two groups for all these parameters (Appendix Fig. S4).

Moreover, we added histology on colon tissue and the liver periportal tracts, which are typically affected by GvHD. There were no signs of GvHD detectable (Fig. EV5). We now mention this on page 11, 2nd paragraph of the Discussion part. We also mention that very recently CRS was observed in CD34-transplanted mice treated with CAR T cells (Norelli et al., 2018; Nat Med advanced online publication) supporting our observation that this is indeed a possible complication developing in these mice (page 12, first paragraph).

(c) Alternatively, the authors could better support their CRS hypothesis if they presented the initial humanization and lymphocyte subset data for the animals and could link CRS to B cell "burden." If M16 and M19 had the highest initial levels of B cells, that might suggest antigen burden was a risk factor for this pathology, as has been observed in clinical trials of CAR T cells.

See response to b). There is no evidence for a correlation with these parameters.

(d) CD34-NSG mice don't have germinal centers. The structures outlined as such in Fig. 3 do not resemble a germinal center, lacking the appropriate density of nuclei and B cells and with no indications of light/dark zones. While clusters slightly enriched for B cell density can be observed in this model, I am not aware of any literature suggesting that these possess the most critical hallmarks of germinal centers. Moreover, the fact that different stains were done on images that are not serial sections further challenges any interpretation of this data.

We are grateful to this reviewer for pointing us to this mistake. We are now using the term B lymphocyte rich zones reminiscent of primordial germinal centers.

(e) In Figures 2E-F, it appears that animals negative for CAR expression by flow and VCN (open symbols) were excluded from analyses of B cell depletion. This cherry-picking of the data is not justified, and it excludes a potentially interesting finding if B cell depletion was also observed in these animals.

We have now added the data for these animals (see revised Fig. 2E+F). Notably, there is no detectable B cell depletion in these mice. B cell depletion is, however, significant for the CAR+ group, also when compared to these mice.

(f) Why is CAR expression so much dimmer in the CD34 model than the PBMC mice? There is a sentence in the text suggesting this is expected, but no reference is cited to support this interpretation. Is this a known characteristic of the SFFV promoter (which is suggested to be used by the reference but not indicated in this manuscript)?

In the CD34 model the CD8 T cells most likely returned to their resting state, since punctual IL-7 stimulation does not last long (3-6 days) (Swainson et al., 2006; J. Immunology 176:6702-6708). In general, promoters are less active in resting cells than in activated T cells. This holds true also for the SFFV promoter in the context of T cells (see Fig. 5B in Frecha et al, 2008; Blood 112, 4843-4852). This is in contrast to PBMC engrafted in NSG mice where especially the CD8 T cells are highly activated by the xenoreactivity. We now refer to this reference (page 7, 2nd paragraph).

(g) The manuscript states that HSC-mice were used "to assess if CAR T cells could also be generated from T cells in steady-state....". However, IL-7 was used in order to activate CD8+ T cells

in the HSC-mice prior to vector injection. Evidence/appropriate controls are needed to show that the IL-7 injections induce proliferation of T cells or increase transduction in vivo.

IL-7 does not induce proliferation of T cells but as a homeostatic cytokine it promotes cell viability and pushes them into the G1B phase of the cell cycle, which makes them more permissive for transduction, especially when administered only temporarily as we did. Our statement "IL-7 promotes T cell expansion" rather refers to continuous use. It was therefore misleading and we have revised this part (page 7, 2nd paragraph). We agree that in absence of a group treated without prior IL-7 injection we cannot demonstrate that IL-7 is crucial in the in vivo approach. However, based on the broad evidence for IL-7 supporting the transduction with lentiviral vectors and also infection of resting T cells with HIV (e.g. Loisel-Meyer et al., 2012, PNAS 109(7):2549-54; Verhoeven et al, 2003, Blood 101: 2167; Cavalieri et al., 2003, Blood 102:497-505) and the fact that the use of IL-7/IL-15 is currently becoming routine in the generation of CAR T cells (Xu et al., 2014 Blood 123:3750-9), it is likely, but will have to be confirmed by future work.

Minor concerns

- Typo on 6th line of the results and discussion: "asses"

The typographical error has been corrected.

- Fig EV 2 looks like its missing some labels. Why are there 3 repeats of PBS/CAR/RFP? Are these the three tissues in order? If so, the data doesn't appear to align with Fig 1G as indicated.

As mentioned above, this figure was labeled wrongly. The three diagrams reflect peritoneum, spleen and blood. We apologize for this mistake which has now been corrected by revising this figure.

- Why do only some histology images have scalar bars? At least 1 should be provided per each tissue.

Scale bars have been included as suggested.

- Figure 1B and 2A need more detail on the graph. Ex: what type of vector is injected and what are the doses were being used. These were explained separately in the legend or method but it is hard for the reader to gather all the information.

We have added the requested information.

- For figure 1D-G, a represented FACS gating was presented. The tissue of the FACS gating should be labeled on the plot.

It is already stated in the legend that peritoneum is shown. The figure is already pretty busy and we feel that adding this information again directly in the figure would make the figure more difficult to understand.

Referee #2 (Comments on Novelty/Model System for Author):

The paper by Pfeffer et al on in vivo generation of CD19 CAR-T cells is of major translational importance for medicine and for your journal. In some ways it is a breakthrough in the field of CAR-T. First was Zelig Eshhar's concept, but while he could technically make them, they were poorly activated in the absence of the appropriate signalling entities. Second was led by June et al adding signalling sequences to CAR which led to fully functional ex-vivo generated CD19 CAR-T, clinically effective.

But at a cost of \$500K per patient! While its not this journals reviewers role to consider the health economics, it is clear that CAR-T are not scalable to all due to problems of manufacture as well as cost. Bucholtz' group here report the first clear steps towards a much simpler scalable and hence cheaper approach generating CAR-T in vivo. This is succinctly and effectively reported. Not only is there efficacy but all the variations and side effects are also present in mice receiving vectors to make CART suggesting what they made is very similar to existing products generated exvivo. The work is well documented and they are very modest about the implications of their work.

This comment is well appreciated.

Thank you for the submission of your revised manuscript to EMBO Molecular Medicine, and my apologies for the unusually long review process. We have now received the enclosed report from the referee that was asked to re-assess it. As you will see the reviewer is now supportive, and I am pleased to inform you that we will be able to accept your manuscript pending the following final amendments:

***** Reviewer's comments *****

Referee #1 (Remarks for Author):

I appreciated the thorough response to my critique

Christian Buchholz
EMBO Mol Med
Manuscript Number: EMM-2018-09158